# Postnatal Growth Assessment of the Very-Low-Birth-Weight Preterm Infant

**DOI:** 10.3390/children12020197

**Published:** 2025-02-06

**Authors:** Kera McNelis, Melissa Thoene, Katie A. Huff, Ting Ting Fu, Zaineh Alja’nini, Sreekanth Viswanathan

**Affiliations:** 1Division of Neonatology, Department of Pediatrics, Emory University, Children’s Healthcare of Atlanta, Atlanta, GA 30322, USA; 2Department of Pediatrics, Division of Neonatology, University of Nebraska Medical Center, Omaha, NE 68198, USA; melissak.thoene@unmc.edu; 3Division of Neonatal-Perinatal Medicine, Department of Pediatrics, Indiana University School of Medicine, Riley Children’s Health, Indianapolis, IN 46202, USA; huffka@iu.edu; 4Division of Neonatology, Perinatal Institute, Cincinnati Children’s Hospital Medical Center, University of Cincinnati College of Medicine, Cincinnati, OH 45229, USA; tingting.fu@cchmc.edu; 5Division of Neonatology, Department of Pediatrics, Mercy Kids Children’s Hospital, University of Missouri School of Medicine, Springfield, MO 65212, USA; zaljani1@mercy.net; 6Nemours Children’s Hospital, University of Central Florida College of Medicine, Orlando, FL 32827, USA; sreekanth.viswanathan@nemours.org

**Keywords:** critical illness, infant, newborn, infant, very low birth weight, intensive care units, neonatal, newborn, premature, premature birth

## Abstract

Preterm birth represents a nutritional emergency and a sudden dissociation of the maternal–placental–fetal unit that regulates metabolic and endocrine physiology. Growth demonstrates health and is a signal of physiological well-being. Growth is expensive for a critically ill infant and possible only after other homeostasis energy demands are met. Despite an expert-stated goal that preterm infants should grow at a similar rate to their gestational age-matched fetal counterparts, this is not the reality for many preterm infants. Other investigators have proposed new metrics for growth quality in the neonatal intensive care unit. This review discusses growth assessment and standards in very-low-birth-weight infants and attempts to address the knowledge gap of which growth metrics are the most important to monitor.

## 1. Introduction

Preterm birth represents a nutritional emergency and a sudden dissociation of the maternal–placental–fetal unit that regulates metabolic and endocrine physiology [1]. Growth demonstrates health and is a signal of physiological well-being. Growth is expensive for a critically ill infant and possible only after other homeostasis energy demands are met. Despite an expert-stated goal that preterm infants should grow at a similar rate to their gestational age-matched fetal counterparts, this is not the reality for many preterm infants, especially those born with a very-low-birth-weight (VLBW) [2].

Meanwhile, experts have proposed metrics for assessing growth quality in the neonatal intensive care unit (NICU). It is a biological phenomenon that individual infants will have growth variability [3]. Although the growth potential of each infant may be different due to their genetic expectation, their concomitant illness, and the ability to provide adequate nutrition to meet metabolic demand, it is still useful for clinicians to have a growth goal in mind. There is significant variability in the approach to growth assessment reporting in both clinical practice and research literature [4]. Therefore, it is worthwhile to improve the understanding of growth during NICU hospitalization and long-term outcomes given the conflicting literature with competing stated thresholds for seemingly optimal growth. This review attempts to address the knowledge gap of which growth metrics are the most important to monitor in the NICU.

## 2. Growth Reference Selection for Very-Low-Birth-Weight Infants

The goal of closely monitoring growth in VLBW infants, like other populations, is to support optimal outcomes and identify the need for alterations in care practices (Table 1). Although as noted by Embleton et al., based on current literature, the ideal growth velocity that optimizes preterm infant outcomes remains unclear [5]. Some expert groups, including the American Academy of Pediatrics (AAP), endorse the goal to strive for preterm infant growth that mimics that of the fetus of the same postmenstrual age [6]. However, others note it is important to consider not only gestational age but also other factors that can influence postnatal growth, including genetics, nutrition, and morbidities [3]. When following either framework, experts generally agree that serial monitoring of postnatal growth is key [3]. There are two types of growth charts used to monitor preterm infant growth: intrauterine and postnatal charts (Table 2) [7,8]. The difference between these two is important when considering their application in clinical practice. Intrauterine, or fetal, growth charts are created using cross-sectional birth anthropometric data for a large cohort of infants over multiple birth gestational ages. Examples of intrauterine growth charts including VLBW infants are Fenton, Olsen, INTERGROWTH-21st, Aris, and Boghossian [9,10,11,12,13]. The use of intrauterine growth references is noted by the AAP to be the best available option at this time to monitor growth in preterm infants [8]. The use of these charts actively applies the concept of targeting growth similar to the fetus. However, one critique of these charts is that preterm infant growth differs from fetal growth, especially in the first few days of life when infants exhibit postnatal diuresis and weight loss but fetuses would continue to gain weight [3,14]. To address this concern, postnatal growth charts are created using longitudinally collected growth data of infants from birth onward. Postnatal standards are thought to represent actual preterm infant growth instead of the ideal growth targeted with intrauterine curves. Two examples of postnatal growth charts including VLBW infants are the INTERGROWTH-21st postnatal chart and the Neonatal Research Network chart [15,16]. One concern with the use of the postnatal charts is the inclusion of infants who are critically ill or have significant morbidities and complications that can alter their growth [5].

When comparing and selecting a growth chart to monitor VLBW infants, multiple factors should be considered. First, not all charts include all anthropometric parameters. For example, the Aris chart only established weight references and Boghossian only established weight and head circumference [9,10]. Additionally, each chart represents variable gestational ages that may not include periviable VLBW infants or includes only a limited number of VLBW infants in general to establish the curves. For example, the INTERGROWTH-21st postnatal growth chart only included 12 infants born between 27 and 32 weeks gestation [16]. Other considerations include the origin of the population sampled to establish the curves (from single or multiple countries) and the features of that population (healthy, all survivors, all births, etc.). It is also important to note that some charts are developed based on completed gestational weeks, which may result in variability and less precision when plotting data (i.e., 23 0/7 weeks plotted same as 23 6/7 weeks). Understanding these differences is key to assist in selecting the most applicable growth chart for a given population and ensuring its correct use.

## 3. Size Classification at Birth

Small for gestational age (SGA) is defined as weight less than the 10th percentile (z-score less than −1.28) at birth. In contrast, large for gestational age (LGA) is defined as birth weight greater than the 90th percentile (z-score greater than 1.28). Growth in SGA infants has been studied with great interest, possibly due to the perception that their nutritional status was negatively altered during pregnancy and that they may need greater nutritional intervention to catch up. Growth in LGA VLBW infants has not been examined as robustly but remains of an area of interest that warrants further investigation.

The frequency of SGA diagnosis is influenced by the growth chart used within a given population. Estañ-Capell et al. noted that the incidence of SGA in a population of infants with median gestational age 30 weeks and birth weight 1290 g varied by growth chart, with the Olsen chart giving a rate of 19.1%, INTERGROWTH-21st 14.2%, and Fenton 10% (*p* < 0.001) [25]. When considering the diagnosis of SGA and the threshold for concern, it is also important to recall that, by design, 10% of the population would be expected to fall into the SGA category. Constitutional smallness is not always equivalent to a higher risk status [3]. The importance of SGA diagnosis in VLBW infants is due to the association of SGA status with an increased rate of neonatal morbidities and mortality, although some have shown that this risk varies by gestational age [26,27]. Additionally, with regard to growth, SGA infants have different growth patterns compared to appropriate for gestational age (AGA) infants, with SGA infants often losing a lower proportion of their birth weight and regaining their birth percentiles sooner [3].

The term SGA should not be used interchangeably with intrauterine growth restriction (IUGR), also called fetal growth restriction. While SGA refers to postnatal size, IUGR refers to intrauterine growth of the fetus. IUGR is a commonly used term in clinical practice but has many definitions in the literature. The American College of Obstetricians and Gynecologists defines IUGR as a fetus with an estimated weight or abdominal circumference less than the 10th percentile [28]. This definition has the same inherent issues in automatically classifying ten percent of a population as “at risk” as a postnatal classification as described above. However, a consensus definition developed by a group using the Delphi method includes a much more extensive definition to differentiate early and late IUGR and includes not only fetal weight and abdominal circumference but also longitudinal growth and umbilical cord Doppler parameters [19]. While a portion of infants who are diagnosed with IUGR also weigh less than the 10th percentile for gestational age at birth and be considered SGA, these terms should not be confused and should not be used interchangeably.

## 4. Weight Monitoring

Initial postnatal weight loss is related to the physiology of contraction of body water compartments, along with the catabolism of endogenous energy stores [29]. Both the degree of prematurity and the fluid and nutrition management can contribute to the magnitude of nadir weight and time to regain birth weight [30]. Both insufficient and excessive early weight loss are associated with a higher risk of mortality [31]. Some high-performing centers recommend that weight is monitored as frequently as every 6 h in the smallest infants [32]. At a minimum, daily weight should be monitored until birth weight is regained [33].

After birth weight is regained or exceeded, daily weights may be recorded, but the assessment of weight gain can occur over a 5–7-day time period to evaluate true growth apart from fluid fluctuations [34]. Extrauterine growth restriction or postnatal growth failure are historical terms referring to an anthropometric measurement under the tenth percentile at 36 weeks corrected age or at discharge [2,18]. However, this concept of a solitary measure of size at 36 weeks corrected age as an independent predictor of neurodevelopment is not supported by current evidence [35]. The growth pattern is more important [36], and the expected weight gain trajectory should be maintained after the postnatal diuresis [3,37,38,39,40]. PediTools.org has a free tool that calculates the weight gain needed over the next week for an individual infant to maintain their growth z-score [41].

The variability of growth velocity calculations can pose a complication in interpreting research findings and adapting into bedside clinical care [42]. There are multiple equations to describe weight changes: the average two-point, exponential two-point, early one-point, and daily methods (Figure 1) [21,43,44]. Variations in calculations create challenges both in bedside clinical application and in interpreting research studies [4]. The early method can over-estimate weight gain if the time period is greater than one week [21]. The average two-point method is recommended by experts to use at the bedside [21]. Generally, weight gain rates of 15–20 g/kg/day using the average or exponential method fit the 23–36 weeks. The growth trajectory can be further clarified by adjusting for the early physiological extracellular fluid loss to create an individualized postnatal target [40,45]. California Perinatal Quality Care Collaborative recommends using a z-score, which measures the distance of a data point from a group mean (Figure 1). The z-score precision is important to display data extremes. Two infants may plot below the first percentile and have the same percentile rank, yet z-scores could demonstrate the magnitude of difference: for example, z-score −3 and z-score −4. Some centers have worked on the visual depiction of z-score and calculation automation in the electronic medical record to aid bedside clinicians in this assessment [46].

## 5. Linear Growth and Ongoing Monitoring

Neonatal length, unlike weight, is unaffected by hydration status. The linear growth velocity of 0.9 to 1.1 cm/week has been reported in preterm infants [47]. Postnatal linear growth as a surrogate of fat-free mass accretion was noted to correlate with brain growth and neurodevelopment [17,48,49,50,51,52,53,54,55]. Postnatal linear growth has been found to be responsive to nutritional interventions [56], especially those affecting protein intake and protein-to-energy ratio, and with enteral vitamin D, zinc and iron supplementation [57,58].

Accurate anthropometry monitoring entails implementing appropriate measurement techniques and interpreting results using appropriate growth charts. Length-board measured neonatal length was shown to be the most reliable and accurate length measurement method [36]. In this measurement technique, the infant is placed supine and fully extended on a rigid length board with knees straightened and feet at the right angle to the body. Two people are needed to perform accurate measurements with one person holding the infant’s head and the other holding the legs [47]. Additional personnel are needed for intubated neonates to avoid accidental extubation. Measurements are recorded to the nearest 0.1 cm.

Trends in linear growth after the second week of life could be used to help guide nutritional management as a part of a comprehensive individualized growth assessment approach that includes nutritional assessment, anthropometric measurement trends, biochemical markers, and clinical history such as prenatal and postnatal morbidities and genetic factors [3,59,60,61]. Low blood urea nitrogen (BUN) values in the context of suboptimal linear growth trend (over 2–4 weeks) in a stable growing preterm infant may be an indication of insufficient protein intake [59]. Implementing an individualized nutritional care approach was noted to help limit postnatal growth deficits [56]. It is important to realize that neonatal anthropometric measurements can have some natural and expected fluctuation and multiple factors may contribute to the linear trajectory. Hence, serial measurements are key to better assess growth patterns [3].

In actual practice, there are barriers to ideal neonatal length measurements, such as the use of nonstandard tools like measuring tape. This is a concern for measurement inaccuracy and the potential for erroneous nutritional interventions [3]. Simple training on the correct two-person techniques using the length board was found to result in accurate measurements with near-perfect intra-class correlation [62]. Finally, inconsistencies in length measurement techniques and the lack of serial measurements across different NICUs could potentially limit the validity of length data in research assessing postnatal growth, its effect on short-term and long-term outcomes, and the effect of various nutrition interventions [56,63].

## 6. Head Circumference Growth and Monitoring

The study by Vasquez-Garibay et al. [64] highlights the significance of head circumference (HC) as a reliable indicator of postnatal growth in VLBW infants. Their research demonstrated that HC measurements correlate well with other growth parameters, offering an effective means to monitor the growth trajectory of these infants. This aligns with the findings from Shoji et al., who reported that the head circumference to chest circumference ratio is a reliable parameter for evaluating fetal growth restriction and predicting physical growth in VLBW children [65]. Multiple studies also indicated that post-discharge head growth is a crucial predictor of improved neurodevelopmental and cognitive outcomes in VLBW infants [66,67]. Ghods et al. found that VLBW infants experiencing HC catch-up growth had better neurodevelopmental outcomes at 5.5 years [66]. Raghuram et al. noted that poor neonatal head growth was associated with significant neurodevelopmental impairment, whereas catch-up growth post-discharge reduced this risk [68]. These findings underline the importance of HC growth as a more robust predictor of neurodevelopmental outcomes compared to other anthropometric measures like weight and length [67]. There is a limitation to its predictive performance. Intraventricular hemorrhage (IVH) is a common complication of prematurity and can result in post-hemorrhagic hydrocephalus [69]. Hydrocephalus can result in an increased HC, but unfortunately, HC is not well correlated with ventricular size measurements [70].

Interventions to enhance HC growth and neurodevelopmental milestones in VLBW infants include enhanced nutrient supply. The study by Strommen et al. demonstrated that an enhanced postnatal nutrient supply improved growth velocity and HC growth, along with better neurodevelopmental outcomes in VLBW infants [71]. The group receiving a nutritional intervention enjoyed significantly improved growth velocity (16.5 vs. 13.8 g/kg/day, *p* = 0.01), increased HC (Δz score: 0.24 vs. −0.12, *p* = 0.15), and decreased mean diffusivity in cerebral white matter areas, such as the superior longitudinal fasciculi, suggesting improved cerebral maturation [71]. Another study by Schneider et al. found that greater energy and lipid intake in the first two weeks of life predicted increased total brain and basal nuclei volumes, as well as improved white matter maturation, which are critical for neurodevelopmental outcomes [72]. Caution with energy provision is important, as another randomized controlled trial found that higher early amino acid provision was associated with lower HC growth [73]. In conclusion, HC growth emerges as a measurable factor in the pathway between nutrition and neurodevelopmental outcomes in VLBW infants [72,74]. The literature consistently supports the use of HC as an early key indicator, with interventions focusing on nutrition and developmental support proving effective in promoting favorable outcomes. However, caution should be used when interpretating HC. A rapid increase in HC due to post-hemorrhagic hydrocephalus is not a sign of healthy growth [75].

## 7. Mid-Upper Arm Circumference

The use of mid-upper arm circumference (MUAC) as a growth indicator in VLBW infants is gaining recognition for its practicality and effectiveness in assessing nutritional status and detecting early growth faltering. A small study including 28 preterm infants found that MUAC correlated with adiposity [76]. Larios-Del Toro et al. observed a significant increase in the percentage of infants with MUAC below −2 z-scores over time, from 23% at the start to 79% by 30 days of hospitalization [77]. This highlights MUAC’s potential sensitivity in detecting early growth faltering, which is critical for implementing timely nutritional interventions aimed at improving outcomes.

Despite these advantages, MUAC should not be used in isolation. Ashton et al. found that while MUAC and mid-thigh circumference have high concordance, they should be used alongside other measures such as weight, length, and HC to provide a comprehensive assessment of growth [78]. MUAC acts as a composite index of inadequate growth, jointly measuring tissue masses and length, which can complicate its interpretation compared to more specific measures like weight-for-length z-scores [79]. Also, MUAC may not be a reliable predictor of regional body composition in preterm infants when compared to measurements obtained by magnetic resonance imaging [80]. Additional information about body composition can be found in another article within this Special Issue, “Body Composition in Preterm Infants: Current Insights and Emerging Perspectives [81]”. In summary, MUAC is an additional marker for the early detection of growth faltering in VLBW infants. Its practicality and reliability make it particularly useful in resource-limited settings, where it serves as a low-cost and effective tool for assessing nutritional status and guiding interventions. However, given the limitations of the reference data and to ensure a comprehensive evaluation of growth and nutritional status, MUAC should be used in conjunction with other anthropometric indicators.

## 8. Body Mass Index

The role of preterm body mass index (BMI) in assessing postnatal growth in VLBW preterm infants is increasingly recognized for its ability to provide a more nuanced understanding of growth patterns. Traditional growth metrics like weight and length may not fully capture the complexity of growth in this population. A study by Williamson et al. emphasized the utility of longitudinal BMI growth charts for preterm infants, which offer a clearer picture of growth trajectories compared to cross-sectional data [82]. These charts reveal distinctive BMI patterns in preterm infants, including a notable nadir followed by gradual increases, which are not captured by traditional metrics such as weight-for-age or length-for-age [82]. Further supporting this, Olsen et al. introduced sex-specific BMI-for-age (g/cm^2^) curves for preterm infants, reinforcing BMI’s role as a measure of body proportionality [24]. Their study highlighted BMI’s stronger correlation with weight than with length, suggesting it as a better indicator of growth status and nutritional needs [24]. Ferguson et al. also validated BMI as a more reliable measure of body proportionality than the ponderal index or weight-for-length index, especially when stratified by gestational age, which is crucial for identifying infants at risk of growth-related complications and tailoring nutritional interventions [83].

However, the use of BMI in this context does have limitations. BMI and other weight-for-length indices can be poor surrogates for adiposity in preterm infants. The studies by Nagel et al. and Ramel et al. showed that these measures do not accurately reflect body composition, particularly fat mass and fat-free mass, leading to significant misclassification of adiposity [84,85]. Moreover, preterm infants exhibit unique growth patterns compared to term infants, often having lower fat-free mass and higher fat percentages at term-equivalent age, complicating the use of BMI for growth assessment [86]. However, accurate body composition measurement methods like air displacement plethysmography have limitations such as cost and operational limitations (e.g., cannot be used for infants on respiratory support) [87]. In clinical practice, integrating BMI into the growth assessment of VLBW preterm infants can aid in the understanding of growth patterns and nutritional needs, potentially leading to better-informed care strategies and improved outcomes. Nonetheless, the limitations of BMI highlight the need for careful interpretation and the integration of more direct measures of body composition when feasible.

## 9. Malnutrition Assessment Through Anthropometry

Malnutrition classification indicators for neonates and preterm infants have been proposed and published by the Academy of Nutrition and Dietetics (AND) as a clinical tool to assess postnatal growth in intensive care and in-patient settings [60]. These suggested malnutrition indicators are intended for use in term infants (born at or after 37 weeks gestational age) up to 30 days of age and preterm infants (born less than 37 weeks gestational age). The criteria are divided into two categories of indicators: the first group of indicators are each standalone and sufficient for a diagnosis of malnutrition, and the second group requires more than one indicator for diagnosis. Malnutrition diagnoses are graded as mild, moderate, or severe.

Although these criteria provide clinicians with an objective assessment of weight gain or linear growth, there are several limitations to their clinical use. First, while they were established by expert consensus, they currently lack validation [61]. For indicators that are based on z-score changes, selection of one of multiple growth charts for preterm infants (e.g., Fenton [11], Olsen [12], or INTERGROWTH [13,16]) for reference may lead to different diagnoses. Studies that examined agreement between the most common preterm reference charts and different definitions of inadequate postnatal growth have reported varying concordance [88,89,90,91,92]. Days to regain birth weight are related not only to natural postnatal diuresis but also daily fluid intake [93], and thus, the AND recommends the use of this metric only in conjunction with nutrient intake. The AND also suggests that weight and length velocities should be compared to individualized goal velocities based on current postmenstrual age and growth percentiles, not commonly accepted targets. While calculating an individualized target likely provides a more accurate assessment, it adds a task to the clinician’s workload.

Furthermore, defining malnutrition based on postnatal indicators alone may not accurately reflect the baseline status of in utero nutrition and growth. SGA infants, which often includes fetal growth restricted infants, typically reach a more attenuated weight loss nadir after birth, are quicker to regain birth weight percentile, and have greater weight gain with more attenuated decline in weight z-score compared to their appropriate for gestational age counterparts [3,94,95]. Based on the malnutrition indicators, this would suggest a lower likelihood of being diagnosed with postnatal neonatal malnutrition. However, they may still be undernourished by other standards [3,96].

## 10. Incorporating Fluid Balance

Careful fluid management is essential for maintaining and promoting the health of a preterm infant, and significant fluid shifts can cloud the ability to assess growth [30]. As mentioned earlier in this article, an initial postnatal weight loss is expected [29]. Weights should be measured frequently but assessed over a 5–7-day time period to evaluate growth, while shorter intervals are more indicative of fluid fluctuations [34]. Strict intake and output should be recorded during the early, acute phase of managing a VLBW infant, and during times of critical illness [33]. Inadequate fluid administration can result in hypovolemia and metabolic derangements. Although fluid restriction is thought to reduce the incidence of patent ductus arteriosus (PDA) in preterm infants, a recent Cochrane review did not find evidence from a randomized controlled trial to evaluate the benefits and harms of fluid restriction for the treatment of symptomatic PDA [97]. There is also low-quality evidence to support fluid administration volume goals in preterm infants with bronchopulmonary dysplasia [98]. This management may be complicated as fluid restriction could restrict the ability to deliver recommended nutrient intake [99]. Excessive daily weight gain may be an indication of volume overload. Serum electrolytes, fluid administration, and losses are incorporated into this determination, and edema is a late sign of fluid accumulation [29].

Fluid accumulation can be trended as follows [29]:


% fluid accumulation = (fluid in − fluid out) ÷ dry weight × 100


## 11. Translating Evidence into Clinical Practice and Common Scenarios

Despite the availability of developed growth definitions, charts, and goals, real-time growth monitoring and nutrition management of VLBW infants require continuous clinical judgment. While the many available tools provide objectivity for clinical assessments, nutrition support and growth management vary between different patients and at differing time points. Practicing clinicians must, therefore, utilize all tools as appropriate yet also employ subjective assessment. This may include the evaluation of available literature, consideration from past clinical experiences, and a nutrition-focused physical exam [60,100,101].

Many examples exist of when clinicians must rely on clinical judgment to evaluate growth or to consider if targeting altered growth goals for a specific infant is more appropriate than standard trajectories. For example, infants who experience altered growth in utero may benefit from postnatal catch-up or catch-down growth to align more with their individualized genetic potential for age. Thus, alternative measures of growth (e.g., linear growth, mid upper arm circumference, etc.) and targeted weight gain goals (e.g., using PediTools.org website [41,102], etc.) can be considered. More complex examples exist, such as VLBW infants with severe chronic lung disease who experience periods of inadequate growth or who receive postnatal corticosteroid treatment and diuretics, which is associated with slower growth [103,104,105]. If linear growth does not meet goal growth trajectories but weight gain does, the clinician needs to decide if this pattern reflects increased adipose tissue and whether this change is acceptable. For these infants who demonstrate slow linear growth trajectories, clinical judgment may, therefore, determine if modifying nutrition support to transiently allow slower weight gain may be appropriate. Likewise, if linear measurements plot lower in growth z-scores for age as compared to weight measurements, clinicians must determine if catch-up growth for weight after a period of inadequate gain is appropriate based on body composition and physical assessment.

More rare instances present the greatest clinical practice challenges, as less objective information is available to provide guidance on growth evaluation and nutrition management. These may include VLBW infants with either indeterminant or diagnosed genetic alterations (e.g., skeletal dysplasia or other syndromes associated with short stature, etc.). These are challenging as postnatal growth goals are altered in term-born affected infants, which may even be available in condition-specific growth charts, yet less information is known regarding the growth of these infants during the fetal/preterm period. These cases may also be complicated by timing of diagnosis; if not known prenatally, postnatal assessments of growth may incorrectly identify their growth as inadequate or meeting suggested criteria for malnutrition, when it may, in fact, be appropriate for their genetic diagnosis [60]. As limited growth data may be available for these children during infancy and childhood, growth must be carefully considered while at VLBW or at preterm gestational ages in order to mimic expected intrauterine growth. Targeted growth goals for these affected infants may be considered based on extrapolation from available published data regarding fetal or birth size at varying gestational ages for affected children.

## 12. Conclusions

With the variety of effects of maternal health conditions during pregnancy, postnatal clinical acuity, medical interventions, and overall medical status on VLBW infant growth, clinicians remain the forefront in managing nutrition support and resulting growth. Consideration of these varying aspects, appropriate use of available growth tools and metrics, and physical assessment of each infant are key for clinical judgment. Only with these comprehensive components can the nutrition and growth of VLBW infants be best supported to promote optimal health and development. Clinicians should use a consistent growth chart, and when possible, use visual depictions of z-scores or plotting measurements on a curve with integration into the electronic medical record. A growth pattern, and not a one-time anthropometric measurement, should be used to determine if a VLBW infant is achieving their potential. In general, most VLBW infants should achieve growth parallel to a centile line on an intrauterine reference chart after their initial expected postnatal weight loss. Additional research is needed to distinguish the mechanisms underlying the association between growth and important long-term outcomes.

## Figures and Tables

**Figure 1 children-12-00197-f001:**
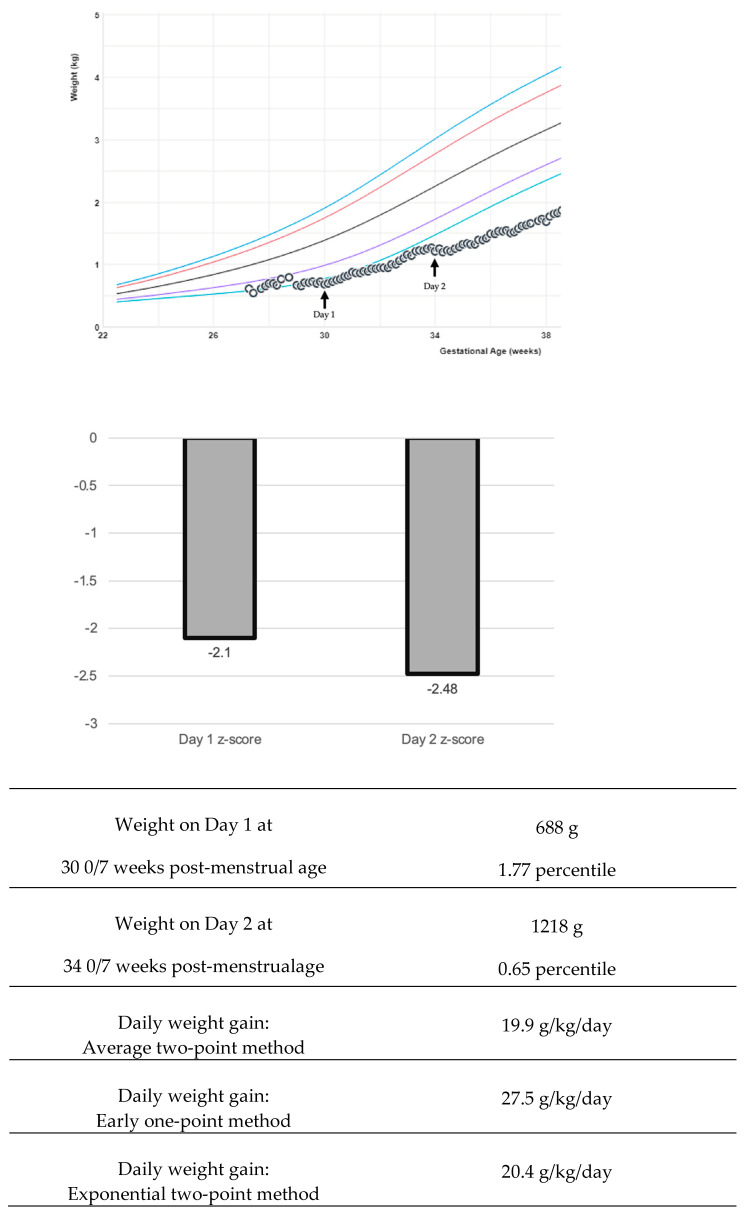
Examples of various growth velocity depictions. In this example, a preterm infant boy’s growth is followed with the Fenton intrauterine growth reference [11,21]. The colored lines in the graph represent the 3rd, 10th, 50th, 90th, and 97th percentiles. Various calculations for weight growth velocity return a different value, and various visual depictions with z-score may be useful to the clinician.

**Table 1 children-12-00197-t001:** Growth monitoring terminology for very-low-birth-weight preterm infants.

Term	Definition	Additional Context or Recommendation
Extrauterine growth restriction	Weight under the 10th percentile at 36 weeks corrected age	Although historically useful, now a solitary measurement is not an independent predictor of neurodevelopmental outcomes [17,18]. This term should no longer be used [18].
Intrauterine growth restriction	Smaller than expected fetal or birth weight following signs of growth deceleration	Fetal weight, abdominal circumference, longitudinal growth, and umbilical cord Doppler parameters are included in this diagnosis [19]. Not all infants that experienced intrauterine growth restriction will be small for gestational age [3].
Intrauterine growth curve	Reference growth chart Built with measurement at birth Reflects fetal/intrauterine growth	Use to determine size at birth Use a consistent growth chart in a newborn intensive care unit [20]
Large for gestational age	Born with weight greater than the 90th percentile	Defined with the use of an intrauterine growth chart
Percentile	The percent of results that would fall below a specified value, related to the median	Used with either intrauterine reference curves or postnatal growth standard charts. May be easier than z-scores for parents to understand.
Postnatal growth standard	Prescriptive standard growth chart Built with measurement over time Typically, smaller sample size and stricter inclusion criteria in the cohort than intrauterine growth curve	Use a consistent growth chart in a newborn intensive care unit [20]
Small for gestational age	Born with weight less than the 10th percentile	Defined with the use of an intrauterine growth chart Not all small for gestational age infants are the product of intrauterine growth restriction [3].
Very low birth weight	Birth weight under 1500 g	
Z-score	Distance from the mean in a normal distribution curve, described in units of standard deviation	Used with either intrauterine reference curves or postnatal growth standard charts. Provides a better illustration of the magnitude of growth faltering.
**Daily weight gain calculation methods expressed in grams/kilogram/day (g/kg/day) over a time period, using weight in grams** [21]		
Average two-point method	$\frac{\frac{\left(weight2-weight1\right)}{\left[\left(weight1+weight2\right)÷2\right]÷1000}}{number\;of\;days}$	Easier to calculate clinically at the bedside
Daily weight gain calculation method	$\frac{\left(weight2-weight1\right)}{\left[\left(weight1+weight2\right)÷2\right]÷1000}$	Daily weights two days in sequence
Early one-point method	$\frac{\left(weight2-weight1\right)÷\left(weight1÷1000\right)}{number\;of\;days}$	Should not be used to summarize weight gain velocity for research studies
Exponential two-point method	$\frac{\ln\left(weight2-weight1\right)\times 1000}{number\;of\;days}$	

**Table 2 children-12-00197-t002:** Comparison of growth charts including very-low-birth-weight infants.

Chart	Birth Gestational Age; Included Gestational Age	Number of Preterm Infants	Unitof Gestational Age	Population	Cohort Characteristics/ Data Used	Chart Type; Anthropometrics Included
Aris [9]	22–42 weeks; 22–42 weeks	61,106 infants 22–33 6/7 weeks	Completed weeks	U.S.A.	Birth certificate data	Intrauterine; Weight
Boghossian [10]	22–29 weeks; 22–29 weeks	133,753 infants	Includes days	U.S.A. and Puerto Rico	VON VLBW Database	Intrauterine; Weight Head circumference
Ehrenkranz (NRN) [15]	N/A *	1660 infants with birth weight 501–1500 g	N/A *	U.S.A.	Infants admitted to NICHD NRN centers at <24 h and lived >7 days	Postnatal; Weight Length Head circumference
Fenton [11] (revised)	22–40 weeks; 22–50 weeks	34,639 infants < 30 weeks	Includes days	Australia, Canda, Germany, Italy, Scotland, U.S.A.	Combined published growth charts; 40–50 weeks using the WHO chart [22]	Intrauterine; Weight Length Head circumference
Intergrowth-21st (very preterm at birth) [13]	24–32 weeks; 24–44 weeks	408 infants	Includes days	Brazil, China, Italy, Kenya, Oman, U.K., U.S.A.	Included some with FGR risks but no ultrasound findings of FGR; 32–44 weeks from Intergrowth-21st original publication [23]	Intrauterine; Weight Length Head circumference
Intergrowth-21st postnatal growth in preterm [16]	27–36 6/7 weeks; 27–64 weeks	28 infants ≤ 33 weeks and 12 at 27–32 weeks	Completed weeks	Brazil, China, Italy, Kenya, Oman, U.K., U.S.A.	Included only accurately dated, uncomplicated pregnancies with well grown fetuses	Postnatal; Weight Length Head circumference
Olsen [12,24]	22–42 weeks; 23–41 weeks	4100 infants ≤28 weeks	Completed weeks	U.S.A.	Pediatrix Medical Group birth data	Intrauterine; Weight Length Head circumference BMI

Of note, additional growth charts for older gestational age preterm or term infants not included in this table. Abbreviations: BMI (body mass index), FGR (fetal growth restriction), NRN (Neonatal Research Network), U.K. (United Kingdom), U.S.A. (United States of America), VON (Vermont Oxford Network), WHO (World Health Organization). * Charted based on birth parameters not gestational age.

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
