# Peer review of "Postnatal Growth Assessment of the Very-Low-Birth-Weight Preterm Infant"

_children, 2025, doi:10.3390/children12020197_

Round 1

Reviewer 1 Report

Comments and Suggestions for Authors

This is a well written review paper on assessments of intra and extrauterine growth in the VLBW preterm infant.  The manuscript could benefit from expansion of some themes, as listed below.  Primarily, the discussion of fluid balance is scattered throughout the manuscript, and it could benefit from its own dedicated section.

Lines 118-121 there is extraneous text

Lines 123-129, please include some discussion of fluid balance, the dangers of fluid overload, because these early changes in weight are important, especially for the extremely preterm infant. Fluid overload has been linked to BPD, need for mechanical ventilation, and NEC.  SGA/IUGR babies may also not have a robust postnatal diuresis, so fluid balance needs to be considered carefully for those infants as well.  This is referenced on lines 297-311 -- and would benefit from bring broken out in its own fluid balance section.

A figure or diagram may be useful to explain the terms in section 4.

Section 6 could benefit from more discussion of HC in the setting of IVH. - as mentioned in lines 224-225

Section 6 - should also caution of aggressive parenteral protein intake and possible negative effects on HC [PMID: 29187120]

Section 7 - more robust discussion of fat free mass (perhaps expanding it into its own paragraph) would add to the manuscript

Lies 331-333 - do infants with BPD receiving diuretics (in addition to steroids) also suffer from worse growth? Is this related to fluid balance?

Author Response

Comment 1

This is a well written review paper on assessments of intra and extrauterine growth in the VLBW preterm infant.  The manuscript could benefit from expansion of some themes, as listed below.  Primarily, the discussion of fluid balance is scattered throughout the manuscript, and it could benefit from its own dedicated section.

Response to comment: Thank you for your suggestion. We agree and have created a new section 10, “Incorporating fluid balance.”

Comment 2

Lines 118-121 there is extraneous text

Response to comment: Thank you for noticing this error. We have remedied the mistake.

Comment 3

Lines 123-129, please include some discussion of fluid balance, the dangers of fluid overload, because these early changes in weight are important, especially for the extremely preterm infant. Fluid overload has been linked to BPD, need for mechanical ventilation, and NEC.  SGA/IUGR babies may also not have a robust postnatal diuresis, so fluid balance needs to be considered carefully for those infants as well.  This is referenced on lines 297-311 -- and would benefit from bring broken out in its own fluid balance section.

Response to comment:  Thank you for your suggestion. We agree and have created a new section 10, “Incorporating fluid balance.”

Comment 4

A figure or diagram may be useful to explain the terms in section 4.

Response to comment:

Thank you for this suggestion. We agree and we have added Figure 1 and Table 1 to explain terminology.

Comment 5

Section 6 could benefit from more discussion of HC in the setting of IVH. - as mentioned in lines 224-225

Response to comment: Thank you for this suggestion. We agree and have added add more to  Section 6.

Comment 6

Section 6 - should also caution of aggressive parenteral protein intake and possible negative effects on HC [PMID: 29187120]

Response to comment: Thank you. We agree and have updated section 6.

Comment 7

Section 7 - more robust discussion of fat free mass (perhaps expanding it into its own paragraph) would add to the manuscript

Response to comment: Thank you for this suggestion. We wholeheartedly agree to the importance of fat-free mass in preterm infants. This is a special issue, and there is an accepted manuscript for this issue dedicated to the topic of body composition of preterm infants. We added a reference to the companion article into this section.

Comment 8

Lies 331-333 - do infants with BPD receiving diuretics (in addition to steroids) also suffer from worse growth? Is this related to fluid balance?

Response to comment:

Thank you for this question. We have updated the sentence to reflect the growth outcomes associated with diuretics. This may be more closely associated with diuretics causing metabolic bone disease than related to fluid balance.

Reviewer 2 Report

Comments and Suggestions for Authors

The manuscript, titled Postnatal growth assessment of the very low birth weight preterm infant, provides an in-depth review of growth assessment strategies for very low birth weight (VLBW) preterm infants. The topic is highly relevant and aligns with ongoing clinical concerns about optimizing outcomes in neonatal intensive care. However, several areas require attention to enhance the manuscript's clarity, scientific rigor, and overall impact.

1.      Abstract: The abstract effectively outlines the study's objectives but lacks sufficient detail regarding key findings or recommendations. Suggestion: Include specific insights or conclusions drawn from the review, such as the strengths and limitations of different growth metrics.

2.      Introduction: Suggest to clearly outline the gaps in existing knowledge and explicitly state how this review addresses them.

3.      While the manuscript discusses multiple growth charts and their applications, it lacks a clear comparison table or visual summary. The manuscript has no visual aids, which could significantly enhance its readability. Add a comparative table summarizing the features, strengths, and limitations of intrauterine and postnatal growth charts. Incorporate figures (e.g., growth trajectories, decision trees) and tables to summarize key findings.

4.      Terms such as "SGA," "IUGR," and "postnatal growth failure" are used extensively but not always clearly differentiated. Provide a glossary or ensure consistent definitions throughout the manuscript.

5.      Offer concrete recommendations for future research or clinical practice in the conclusion.

Author Response

Reviewer 2

The manuscript, titled Postnatal growth assessment of the very low birth weight preterm infant, provides an in-depth review of growth assessment strategies for very low birth weight (VLBW) preterm infants. The topic is highly relevant and aligns with ongoing clinical concerns about optimizing outcomes in neonatal intensive care. However, several areas require attention to enhance the manuscript's clarity, scientific rigor, and overall impact.

  1. Abstract: The abstract effectively outlines the study's objectives but lacks sufficient detail regarding key findings or recommendations. Suggestion: Include specific insights or conclusions drawn from the review, such as the strengths and limitations of different growth metrics.

Response: Thank you for the suggestion. We agree and have added additional text into the abstract.

  1. Introduction: Suggest to clearly outline the gaps in existing knowledge and explicitly state how this review addresses them.

Response: Thank you for the suggestion. We agree and have added additional text into the Introduction.

  1. While the manuscript discusses multiple growth charts and their applications, it lacks a clear comparison table or visual summary. The manuscript has no visual aids, which could significantly enhance its readability. Add a comparative table summarizing the features, strengths, and limitations of intrauterine and postnatal growth charts. Incorporate figures (e.g., growth trajectories, decision trees) and tables to summarize key findings.

Response: Thank you for the suggestions. We agree and have added Figure 1 and Tables 1 and 2.

  1. Terms such as "SGA," "IUGR," and "postnatal growth failure" are used extensively but not always clearly differentiated. Provide a glossary or ensure consistent definitions throughout the manuscript.

Response: Thank you. We agree and have added Table 1 with these definitions.

  1. Offer concrete recommendations for future research or clinical practice in the conclusion.

Response: Thank you for the suggestion. We agree and have added additional text to the Conclusion.